# Mask-Pyramid Network: A Novel Panoptic Segmentation Method

**DOI:** 10.3390/s24051411

**Published:** 2024-02-22

**Authors:** Peng-Fei Xian, Lai-Man Po, Jing-Jing Xiong, Yu-Zhi Zhao, Wing-Yin Yu, Kwok-Wai Cheung

**Affiliations:** 1Department of Electronic Engineering, City University of Hong Kong, Hong Kong; xian.pf@my.cityu.edu.hk (P.-F.X.); jingxiong9-c@my.cityu.edu.hk (J.-J.X.); yzzhao2-c@my.cityu.edu.hk (Y.-Z.Z.); wingyinyu8-c@my.cityu.edu.hk (W.-Y.Y.); 2School of Communication, The Hang Seng University of Hong Kong, Hong Kong; keithcheung@hsu.edu.hk

**Keywords:** panoptic segmentation, convolutional neural network, image processing

## Abstract

In this paper, we introduce a novel panoptic segmentation method called the Mask-Pyramid Network. Existing Mask RCNN-based methods first generate a large number of box proposals and then filter them at each feature level, which requires a lot of computational resources, while most of the box proposals are suppressed and discarded in the Non-Maximum Suppression process. Additionally, for panoptic segmentation, it is a problem to properly fuse the semantic segmentation results with the Mask RCNN-produced instance segmentation results. To address these issues, we propose a new mask pyramid mechanism to distinguish objects and generate much fewer proposals by referring to existing segmented masks, so as to reduce computing resource consumption. The Mask-Pyramid Network generates object proposals and predicts masks from larger to smaller sizes. It records the pixel area occupied by the larger object masks, and then only generates proposals on the unoccupied areas. Each object mask is represented as a H × W × 1 logit, which fits well in format with the semantic segmentation logits. By applying SoftMax to the concatenated semantic and instance segmentation logits, it is easy and natural to fuse both segmentation results. We empirically demonstrate that the proposed Mask-Pyramid Network achieves comparable accuracy performance on the Cityscapes and COCO datasets. Furthermore, we demonstrate the computational efficiency of the proposed method and obtain competitive results.

## 1. Introduction

Benefiting from further exploration of deep convolutional neural networks, panoptic segmentation [1] has been a long-standing and practical research topic in the field of computer vision. This task focuses on combining two challenging tasks, semantic segmentation [2] and instance segmentation [3]. In the beginning, semantic segmentation aimed to predict a semantic label for each pixel of an image. However, it cannot distinguish between different objects of the same class. For example, a crowd of persons is marked with the same “person” label. Instance segmentation distinguishes different objects but misses uncountable backgrounds. Some classes have separable objects denoted as “things”, such as cars, and persons, while other classes are hard to separate into several objects and are denoted as “stuff”, such as sky, and road ground. Instance segmentation can only detect “things” but leaves unlabeled pixels of “stuff”. The motivation of this work is to solve the drawbacks, reserve the strength of both segmentation tasks, and fuse the two tasks in a unified model architecture.

Based on the above motivation, panoptic segmentation is proposed. Since the panoptic segmentation task was proposed by Kirillov et al. [1], it has become an intuitive and popular solution to extend the well-known instance segmentation network Mask RCNN [4] by adding a semantic segmentation branch and fusing the outputs. The Mask RCNN family has the merit of good generalization and has been widely used in image classification, object detection, and instance segmentation tasks. Credited to its good performance, the Mask RCNN family has formed a huge community, accumulating a large number of optimization algorithms and model parameter resources. De Geus et al. [5] introduced Mask RCNN to the panoptic segmentation task for the first time and proposed JSIS-Net, which added a Pyramid Pooling Module as a semantic branch and merged “things” and “stuff” using heuristics. However, the performance of JSIS-Net is poor. Xiong et al. [6] optimized it and proposed UPSNet, which introduced a panoptic head to resolve conflicts when fusing “things” and “stuff”. UPSNet achieves good performance, but there are still many problems in panoptic segmentation based on Mask RCNN.

The problems of conventional methods based on Mask RCNN can be summarized into three points:(1)The Mask RCNN family methods are usually inefficient when densely generating too many boxes, which is a heavy burden for successor post-processing and inference stages. As the basis of Mask RCNN, Ren et al. [7] densely generated a large number of boxes as object proposals, most of which were discarded in the post-processing stage. On average, three box proposals are generated per pixel of each level feature map, and then most of them are suppressed by the few valid instance masks. Generating these boxes, as well as the succeeded score regression and mask prediction process, consumes significant computational resources. The reason for generating so many object proposals is to better handle small-scale objects, but UPSNet still performs poorly on small objects.(2)Merging semantic segmentation and instance segmentation outputs is a challenge for Mask RCNN-based methods. From the heuristic fusion of JSIS-Net to the panoptic head of UPSNet, merging “things” and “stuff” is the bottleneck to unlocking the potential of network design. The fusion module needs to solve the overlapping problem of instance masks. Furthermore, since the semantic segmentation output does not distinguish different “things”, the outputs of the two subtasks cannot be directly compared, making it difficult to fuse the outputs of the semantic and instance segmentation subtasks.(3)In Mask RCNN-based panoptic segmentation, the information flow is cut into 2–3 stages, blocking out the end-to-end optimization of backpropagation. Because, in the intermediate stage, the box object proposals are represented by a four-number tuple from regression, however, the network features at the backbone stage and the panoptic segmentation masks at the head stage are represented in *H* × *W* × *C* spatial format, features are repeatedly converted and cropped in the pipeline. Frequently changing the information format will degrade the performance of the neural network.

In response to these problems, some works abandon the Mask RCNN’s idea of generating box object proposals but turn to exploring some box-free approaches. Cheng et al. [8] proposed Panoptic-DeepLab based on the well-known semantic segmentation network DeepLab [9,10,11], which regresses instance centers and offsets bias for each pixel to the corresponding center. Wang et al. [12] proposed PCV based on Hough Transform, which performs discrete probabilistic voting on possible pixels to generate instance centroids. Li et al. [13] proposed Panoptic FCN to encode each object instance or stuff category into a specific kernel weight using the proposed kernel generator and produce predictions by directly convolving high-resolution features. These box-free methods provide us with more imaginative ideas, however, these box-free ideas do not perform as well as the Mask RCNN-based methods. The Mask RCNN family methods can provide more accurate and robust panoptic segmentation results.

To address the above problems and take advantage of the high precision of the Mask RCNN family, we propose a novel method called the Mask-Pyramid Network (Mask-PNet). The basic idea is to avoid generating too many proposals by exploiting the mutual interaction between object proposals. Since most box object proposals are covered and suppressed by several high-score masks, we can utilize the first detected object masks to avoid generating meaningless proposals under the coverage area of the first detected object masks. The proposed Mask-PNet first detects larger objects and then predicts the mask logits of these large objects through a simple feature pyramid network [14]. Once the larger objects and affiliated masks are obtained, there is no need to generate proposals for pixels under the predicted mask. The network then focuses on uncovered regions to detect middle-scale and small-scale objects. The above process is repeated, objects from large scale to small scale are detected, and masks are obtained until all the uncovered regions are filled with object masks. During the process, the Mask-PNet generates far fewer object proposals. After this process, we can obtain the mask logits of all the mask proposals. By applying SoftMax, we can obtain the final masks. Both semantic and instance segmentation results are placed in a unified mask logits tensor, where “things” and “stuff” merge naturally. During the training of the network, by applying a cross-entropy loss, the gradient can be directly backpropagated to the semantic and instance segmentation branches simultaneously.

In practice, the pyramid scheme refers to the feature pyramid [15]. The feature pyramid is the output of backbones such as ResNet [16] and HRNet [17]. The bottom layer of the feature pyramid has a 128*128 size, while the top layer is as narrow as 2 × 2 pixels, which means, in the first step, we only need to generate four proposals for the larger objects at the top feature layer. These top-layer proposals have the largest perceptron area and usually predict the largest object masks in the image. After predicting the mask for the top-level feature layer pixels, the second top-level layer is processed in the same way. At lower layers, pixels not covered by existing masks are allowed to generate new proposals and predict their masks. This proposal-generating process is repeated level by level until the bottom feature layer is processed. Generally speaking, employing the pyramid scheme can reduce the number of proposals.

Despite generating fewer proposals, the Mask-PNet achieves competitive results on the Cityscapes panoptic segmentation benchmark. Experiments show that the proposed method outperforms the mainstream methods by 2% while greatly reducing the consumption of computing resources. Overall, the main contributions of this paper can be summarized as:This paper provides a novel algorithm to significantly reduce the number of object proposals, by properly managing the generation process of object proposals.This paper proposes to integrate the mask logits of semantic segmentation and instance segmentation into a unified tensor of mask logits, providing a novel approach for merging the outcomes of these two subtasks.This paper proposes a novel joint training mechanism that enables the interaction of gradients and loss information between the semantic and instance segmentation subtasks.The experimental results illustrate that the proposed Mask-PNet achieves superior performance compared to mainstream panoptic segmentation techniques while exhibiting a significant reduction in computational requirements.

## 2. Related Works

Panoptic segmentation was introduced in [1] to unify the tasks of semantic segmentation and instance segmentation, which has been quite studied in recent years due to the rapid development of the Convolutional Neural Network (CNN) [18]. Since most panoptic segmentation methods are modified from these two subtasks, we introduce the technical routes for semantic segmentation and instance segmentation since CNN was employed.

Semantic Segmentation: Semantic segmentation has been one of the most extensively studied computer vision tasks in recent years. Many of the state-of-the-art methods are built upon Fully Convolutional Networks (FCNs) [19], an end-to-end encoder–decoder architecture. The encoder of the FCN is based on the VGG-16 architecture [20], where the inner product layers are replaced by convolution layers. This enables the network to produce pixel-wise segmentation maps of the input image in a single forward pass. SegNet [21] introduced unpooling layers as an alternative to transposed convolutions for upsampling high-level features. This reduces the number of parameters in the network, making it more computationally efficient. PSPNet [2] proposed pyramid pooling, which enables the network to consider features at different scales. It incorporates multi-scale features and constructs a pyramid of pooling layers with different dilation rates. This allows the network to aggregate information from different spatial scales, improving the overall segmentation performance. Yu and Koltun [22] introduced atrous convolution, also known as dilated convolution, to expand the receptive field without increasing the number of parameters by enlarging the coverage of filter kernels. The Atrous Spatial Pyramid Pooling (ASPP) series [10,23,24] concatenates features from atrous convolutions with different dilation rates via global pooling [25]. This enhances the performance of semantic segmentation by allowing the network to capture both global and local context information. The state-of-the-art semantic segmentation methods are built on top of FCN-based architectures, which have been extended with various techniques, such as unpooling, pyramid pooling, and atrous convolution. These techniques have significantly improved the accuracy of semantic segmentation while reducing the computational complexity of the networks.

Instance Segmentation: Instance segmentation is a task in computer vision that involves detecting and segmenting individual objects within an image. In recent years, there has been significant progress in instance segmentation, driven by the development of deep convolutional neural networks. Dai et al. [26] proposed a three-network design, for instance, which included segmentation, which includes sub-networks for instance detection, mask prediction, and object classification. In this approach, the instance detection sub-network generates proposals for object instances, which are then refined by the mask prediction sub-network to generate pixel-level masks. The object classification sub-network predicts the class label of each object instance. Mask R-CNN [4] is a popular box-based approach for instance segmentation that extends Faster R-CNN [7], a widely used object detection method. Mask R-CNN constructs a shared backbone and predicts three head sub-branches in parallel, including bounding box regression, classification, and mask prediction. This approach has achieved state-of-the-art results on several instance segmentation benchmarks and has set a trend in, for instance, segmentation-related tasks. Liu et al. [27] improved Mask R-CNN by introducing path augmentation to enhance object localization in the backbone. The authors added auxiliary connections to the backbone network, which allowed the network to learn richer feature representations. This approach improved the accuracy of object detection and instance segmentation, particularly for small objects. Instance segmentation has seen significant progress in recent years due to the development of deep neural networks and the introduction of innovative approaches, such as Mask R-CNN and path augmentation. These methods have enabled the accurate detection and segmentation of individual objects within complex scenes and have broadened the range of applications for instance segmentation in computer vision.

Panoptic Segmentation: Panoptic segmentation is a recently introduced framework that integrates both semantic segmentation and instance segmentation tasks. In recent years, panoptic segmentation methods have seen significant advancements due to the rapid development of Convolutional Neural Networks (CNNs). Box-based methods have been widely adopted in most panoptic segmentation approaches. Mask R-CNN, a popular box-based method, extends Faster R-CNN to predict three head subbranches in parallel, including bounding box regression, classification, and mask prediction. Panoptic FPN [13] takes inspiration from Mask R-CNN and adds a semantic segmentation head to predict stuff regions from the feature pyramid. AUNet [28] improves the fusion of the instance and semantic branches by introducing an attention mechanism into Mask R-CNN. UPSNet [6] proposes a parameter-free head for thing–stuff fusion. The Spatial Ranking module, proposed by Liu et al. [29], aims to separate things and stuff and improve the feature representation of the instance branch. Li et al. [30] proposed learning a binary mask to explicitly constrain output distributions of stuff and things. While most state-of-the-art semantic segmentation methods are built upon Fully Convolutional Networks (FCNs), instance segmentation methods usually adopt a box-based approach. More novel attempts have been made to improve the performance of panoptic segmentation, and research in this field is ongoing.

## 3. Mask-Pyramid Network

The overview of the Mask-PNet architecture is illustrated in Figure 1. There are two branches: the semantic segmentation (SS) branch and the Mask Pyramid (MP) branch. The semantic segmentation branch is a typical U-Net style network, as shown in the upper branch in Figure 1. The Mask Pyramid branch, shown in the lower area of Figure 1, is used to separate objects from each other and to distinguish objects from the same category. By concatenating the stuff part of the semantic segmentation result with the k-object Mask-PNet instance segmentation result, we can obtain the final logits and panoptic segmentation masks. In this way, the k-object mask logits and the stuff mask logits are combined together to conduct the cross-entropy loss, the loss function can directly calculate the gradients based on the overview of the semantic segmentation branch and the mask pyramid branch. This provides a novel joint training mechanism that enables the interaction of gradients and loss information between the semantic and instance segmentation subtasks.

### 3.1. Semantic Segmentation Branch

The semantic segmentation (SS) branch is used to generate coarse semantic segmentation results as a basic reference for Mask-PNet. The main modules of this branch contain the backbone and the semantic segmentation sub-network. The backbone network extracts spatial features and down-samples feature sizes level by level. Most of the mainstream deep learning networks, like ResNet, ResNeXt, and HRNet, can be employed as the backbone. The semantic segmentation sub-network reassembles the features and progressively upsamples the feature sizes to one-fourth of the original input image size. Denoting the original input image size as H×W×3, the semantic branch output is a 1/4H×1/4W×C, $\left(C=C_{1}+C_{2}\right)$ size tensor, where $C_{1}$ denotes the number of stuff categories and $C_{2}$ denotes the object number of things. This is a typical cross-entropy approach to organize category logits in semantic segmentation. The semantic segmentation branch affects the final results in two ways. On the one hand, the coarse segmentation result of the semantic branch is an important basis for dividing things and stuff. On the other hand, the logits of stuff in the semantic segmentation result will be concatenated together with the logits of the instance object, so as to conduct SoftMax for producing the final segmentation result.

### 3.2. Mask Pyramid Branch

The Mask Pyramid branch (MP) is used to generate mask pyramid proposals and compute mask logits for each object in each image. Benefiting from the following unique model algorithms, Mask Pyramid can suppress the number of generated proposals and reduce the consumption of computing resources. The MP branch shares the backbone with the semantic branch. Over the top of the backbone, we deploy a keep-down sub-network to further reduce the feature size and enlarge the receptive field. When we combine the backbone features and the keep-down features, then feed them into the Mask Pyramid’s module, we can obtain a 1/4H×1/4W×k output tensor, where k is the number of detected objects from the things category. As mentioned before, our approach detects objects and computes their masks in a box-free manner, which is implemented in this branch.

The MP branch also uses the features from the typical ResNet backbone, these features contain various sizes and receptive fields. In the case of 512 × 512 size input images, the backbone extracts feature at spatial resolutions of 1/4, 1/8, 1/16, and 1/32. In order to obtain a larger receptive field and facilitate the generation of mask pyramids, we introduce a keep-down ResNet module to obtain features at 1/64, 1/128, and 1/256, so the top feature layer has a spatial resolution of 2 × 2 and the entire feature extractor (backbone + keep-down) looks like a pyramid with a steeple top. As is shown in Figure 2, the MP branch is a pipeline to produce objects in a feature–pyramid architecture. Figure 2 consists of two parts, the left part (a) demonstrates the overall Mask-PNet Pipeline, and the right part (b) illustrates, in detail, how features and seed maps work together to generate the object proposals with high efficiency.

### 3.3. Mask-Pyramid Network Pipeline

The overview of the Mask-Pyramid Network pipeline is demonstrated in the left part (a) of Figure 2. We first obtain the features from the ResNet backbone and keep-down module with spatial sizes from 128×128 to 2×2. From the Semantic segmentation branch, we obtain the semantic segmentation logits, with a size of $128\times 128\times \left(C_{1}+C_{2}\right)$. Next, we maintain a $128\times 128\times \left(C_{1}+k\right)$ updating logits to record the predicted stuff and things, where $C_{1}$, $C_{2}$, and k refer to the stuff category number, the things category number, and the detected object proposals number. In the updating logits, it includes the stuff category number $C_{1}$ and the detected object proposals number k but excludes the things category number $C_{2}$, because the things category is represented by the more fine-grained object proposals. So, our initial logits come from the stuff logits of the semantic segmentation branch outputs.

Then, we can generate a seed map based on the initial logits. The mask area of each category in SS output is obtained by an argmax operation, the initial logits at the things area should be left blank, and we introduce a seed map to record these empty areas. The empty area has a size of 128×128, and we use an average low-pass filter to reduce the seed map size to match the features, such as 2×2, 4×4, etc. We use this seed map to control the proposal-generating process of features. Only those feature pixels with a positive seed map value can generate an object proposal. In the implementation, we directly multiply the features with the seed map, we obtain the filtered features, and then generate object proposals as Mask RCNN. These new proposals are processed by a convolutional neural network (CNN) to produce new logits. By concatenating the existing logits and the new logits, we obtain the updated logits. This detailed logits-updating scheme is illustrated in the right part (b) of Figure 2.

In short, we can obtain a 2×2 seed map from the existing logits and use this seed map to filter the 2×2 features and control the number of newly detected proposals. These new proposals are fed into CNN to produce new logits and update the logits. The new logits cover part of the empty area, then we can obtain a new 4×4 seed map. Analogously, the Mask-PNet produces 4×4 filtered features and generates update logits with smaller receptive fields. We repeat this process to generate object proposals on 8×8, 16×16, 32×32, 64×64, 128×128 size features, until the seed map is fully negative, or all the feature levels are used. In this way, we can generate object proposals and predict their logits and masks in order, from the larger to the smaller sizes.

Similar to the Mask RCNN [4] and regional proposal network [3], Mask-PNet also needs to generate object proposals. The proposal-generating process of Mask-PNet is inspired by these approaches but has two differences from them. Mask RCNN generates three proposals on each pixel of the feature map from every four feature levels, then it predicts a four-element tuple to draw a box on the feature map, and masks are inferenced within the box area. Firstly, Mask-PNet generates much fewer proposals, by introducing the seed map our method avoids consuming resources on those feature pixels that are meant to be suppressed by larger objects. Secondly, the CNN of Mask-PNet can utilize the full feature map for generating new logits and masks, however, Mask RCNN can only use the bound-box cropped features for predicting the masks.

Mask-PNet is trained to detect as many objects as possible in an image. Some object proposals can correctly hit objects and will predict masks according to the features, while some object proposals do not hit any object, and a blank area will be left during the mask prediction process. These mask logits interact with each other in the proposal-generating scheme. The instantiation of mask logits is managed by the seed map, which will be discussed in detail in the next section. The size of the seed map is the same as the feature size at each level. As the seed map works down the feature pyramid level by level, more and more mask logits are instantiated. In the Mask-PNet, large-area objects are first detected and instantiated. Note that each mask proposal produces an exclusive layer of mask logits, each value representing how many of the corresponding pixels belong to that object. For objects occupying similar area sizes, objects with higher category scores in the semantic segmentation logits will be instantiated first. Finally, we obtain k object proposals and compute their k mask logit layers. These mask logits are upsampled to the original input image size and combined with $C_{1}$ stuff mask logits layers, which then emit $H\times W\times \left(C_{1}+k\right)$ mask logits. By applying the argmax operation on the channel dimension, we can obtain the final panoptic segmentation results from the maximum indices.

### 3.4. Seed Map Updating

The relationship between the seed map and the new mask pyramid is mutually interactive. The seed map is designed to control the mask proposal generating process, also, the generated mask logits update the seed map at the next level. The seed map is calculated based on the existing mask logits and proper feature size. Firstly, we calculate the 128×128 empty area based on the existing mask logits. The empty areas are bool values, positive for empty and negative for occupied. This value depends on whether the thing logit is larger than the stuff logit or existing mask logit at each pixel. Secondly, the size of the seed map should be the same as the feature size of the corresponding feature level. Mask-PNet uses the average low-pass filter to down-sample the empty area into a suitable size while reserving the occupation information at each level and marking values over 0.5 as positive. In practice, we employ the area interpolation operation of PyTorch. Note that, in pixels where things logits are greater than stuff logits and the existing mask pyramids logits are set to positive it means that a new object should be generated here, while other pixels are set to negative.

New object proposals are generated according to the seed map. Pixel with the highest thing logits response will generate the new mask logits first. A new object proposal in turn computes its mask logits and participates in the next level generating of the seed map, because of the new mask logits, the thing logits are no longer maximal at some pixels. Repeating this “seed map—mask logits—seed map” generating process, new mask logits are generated one by one until the seed map is completely negative. The mask logits updating process is illustrated in Figure 3.

### 3.5. Mask Pyramid Instance Generating Algorithm

We can summarize the process of the Mask-PNet mask logits updating process into a concise algorithm, as shown in Algorithm 1. For the sub-network, one of the inputs is down-sampled features feat with sizes of 128, 64, 32, 16, 8, 4, and 2, where *H* = *W* = 128 represents the mask logits size. Another input is the coarse semantic segmentation result, which is a tensor of shape $H\times W\times (C_{1}+C_{2})$.


**Algorithm 1** Mask-PNet Mask Logits Updating
*/* For better understanding, some tensor sizes are as below*/*

*/* semantic: [*

$H,\;W,C_{1}+C_{2}$

*], stuff: [*

$H,\;W,C_{1}$

*], things: [*

$H,\;W,C_{2}$

*]*/*

*/* logits: [*

$H,\;W,C_{1}+k$

*], empty: [*

H, W

*], seed_map: [h, w]*/*

*/* new_logits: [*

$H,\;W,k_{i}$

*], feat:[h, w, c], filtered_feat:[h,w, c]*/*

*/* proposals: [*

$k_{i},\;c$

*], where H = W = 128 */*
        **Input:** Features feat with sizes in *[(128,128), (64, 64), …, (2,2)]*;                    Semantic segmentation results: stuff and things.        **Output:** Panoptic Mask Logits: logits.    1    logits← stuff;    2      **for** i=  1 to  7 **do**    3                    h, w←sizesi;    4                    empty← things>logits;    5                    seed_map←AvgLowPassempty, h,w>0.5;    6                    **if**    seed_map.sum >0 **do**    7                                    **break**;    8                    filtered_feat← feat*seed_map;    9                    proposals =MaskPyramidfiltered_feat;  10                      new_logits =CNNfeat, proposals;  11                      logits← Concatenatelogits, new_logits;*                                            /* Return panoptic segmentation mask logits.*/*  12        **return** logits;


In order to facilitate the calculation and comparison of thing logits, we compress the stuff logits and the mask pyramids logits into panoptic mask logits with a shape of $H\times W\times (C_{1}+k)$. Mask logits are designed to collect stuff logits and new running mask logits, and so initialize with stuff logits, and at first, the number of new mask pyramids to k=0.

Next, we generate mask logits and update the seed map level by level. As mentioned before, the seed map is determined by comparing the thing logits with the panoptic logits. At each feature size scale, the empty area is down-sampled with the average low-pass filter to the same size to obtain the corresponding seed map. The values of the seed map are 0 and 1. One means that the things pixel value is larger than the value of the current panoptic logits. As is expressed in Equation (1), the pixel value of the seed map at the coordinate $\left(i,\;j\right)\;seed\_map_{ij}$, is marked as 1 if the corresponding value of the things map is larger than the logits value, otherwise, it will be marked as 0.
(1)$$seed\_map_{ij}=\left\{\begin{array}{c} 1,\;if\;thing>logits\\ 0,\;others \end{array}\right.\;$$

Then, we repeatedly generate a new mask logit and update the seed map until the seed map is turned to 0 at every pixel. In order to generate the new mask logit, we need to obtain an initial position guided by the seed map. The principle is to select the pixel position with the largest thing logits, and the seed map is positive. So, before the argmax operation, we perform an element-wise multiplication between the logit of the thing and the seed map, which indicates the ideal position for the new mask logits.

Now, we compute the new mask logit in a cross-entropy style neural network output. The pixel at the current seed map is turned negative. Also, the number of mask pyramids *k* increases. The new mask is defined as a matrix map where the new mask logit is larger than the existing panoptic logits. The new mask logit is then concatenated into panoptic logits. The seed map is also updated by turning negative on the pixels covered by the new mask. After turning off each pixel of the seed map for each size level, we obtain the final accumulative panoptic logits.

### 3.6. Fusion and Loss Function

The fusion process of the Mask-PNet is straightforward. For the Mask RCNN-based panoptic methods, instance predictions are required to match positions and replace the semantic predictions in a difficult way. Our method inherits the idea of cross entropy, which is to determine the prediction of each pixel by comparing the logit values of each class or instance. For cross-entropy style prediction, the basic requirement is to concatenate the stuff logits and instance logits. In fact, our Mask-PNet already performs this process during mask logits generation, and the stuff prediction and instance predictions have been fused into panoptic logits naturally already.

Per-pixel cross-entropy loss [31] is popular when training neural networks to solve segmentation problems. Our Mask-PNet also employs per-pixel cross-entropy loss to tune model parameters. There are two loss functions in our model, the loss for the semantic segmentation branch and the loss for the final fusion panoptic segmentation result. Both loss functions can be summarized as
(2)$$L=\sum_{i}\left(y_{i}\log p_{i}\right)$$
where i refers to each pixel, $p_{i}$ refers to the pixel of the predicted mask, $y_{i}$ refers to the pixel of the ground truth label. Because the logits of stuff and things are integrated into a unified tensor by MPN, which has the same tensor shape as the traditional semantic segmentation results, we can directly use the widely used typical per-pixel cross-entropy loss.

The semantic segmentation loss function and fusion panoptic segmentation have different numbers of categories. For the semantic segmentation loss, each category refers to the semantic class number, i.e., the sum of the stuff classes and the thing classes. Since the categories are predefined, the category number is always fixed. For the panoptic segmentation loss, the category consists of the stuff classes and thing instances. The number of the stuff classes is fixed, while the number of thing instances varies from image to image, depending on how many objects the image contains.

## 4. Experimental Results

In this section, we first introduce the typical metrics we used for evaluations, and then briefly describe the dataset used in our experiments. We then present extensive quantitative comparisons and detailed ablation studies on the proposed architectural components. Finally, we present a visualization of a panoptic segmentation evaluation of the dataset.

### 4.1. Experiment Datasets and Evaluation Metrics

In our experiments, the public panoptic segmentation datasets Cityscapes [32] and COCO [33] are employed. The Cityscapes dataset consists of urban street scenes and focuses on a semantic understanding of common driving scenarios. Due to its diversity, it is one of the most challenging datasets for panoptic segmentation, as it covers scenes recorded in different seasons (such as spring, summer, and autumn) from more than 50 European cities. The existence of a large number of dynamic objects further increases its complexity. The scenes in the Cityscapes dataset are extremely cluttered and contain many dynamic objects, such as pedestrians and cyclists, that are often close to each other or partially occluded. These factors make panoptic segmentation, especially the ‘thing’ classes, more challenging.

We evaluate the proposed Mask-PNet by comparing it with other well-known architectures based on the evaluation metric of Panoptic Quality (PQ) [1], which is defined as
(3)$$\mathrm{PQ}=\frac{\sum_{\left(p,\;g\right)\in TP}IoU\left(p,\;g\right)}{\left|TP\right|+\frac{1}{2}\left|FP\right|+\frac{1}{2}\left|FN\right|}$$
where *p* and *q* refer to the predicted mask region and the ground-truth mask region, respectively. TP, FP, FN, and IoU are true positives, false positives, false negatives, and intersection-over-union, respectively. We calculate IoU as $IoU=TP/\left(TP+FP+FN\right)$. We also report Semantic Quality (SQ) and Recognition Quality (RQ) metrics, calculated as
(4)$$\mathrm{SQ}=\frac{\sum_{\left(p,\;g\right)\in TP}IoU\left(p,\;g\right)}{\left|TP\right|}$$
(5)$$\mathrm{RQ}=\frac{\left|TP\right|}{\left|TP\right|+\frac{1}{2}\left|FP\right|+\frac{1}{2}\left|FN\right|}$$

Note that PQ is the product of semantic quality and recognition quality. SQ refers to the mean Intersection over Union (mIoU) of the true positive mask pyramids, which indicates how accurately the masks are predicted. RQ calculates the recall rate, which reveals how many instances can be identified rather than missed. We report PQ, SQ, and RQ for all the classes in the dataset, we also report them for ‘stuff’ classes (PQ^St^, SQ^St^, RQ^St^) and the ‘thing’ classes (PQ^Th^, SQ^Th^, RQ^Th^). Furthermore, to better compare the efficiency of the proposed Mask-PNet and other traditional semantic segmentation and instance segmentation models, we also report the mIoU for “stuff” and “thing” classes and infer GPU memory and FLOPs for comparison. We use PyTorch V1.12 [33] to implement our architecture and train the proposed model on a single GPU.

### 4.2. Ablation Study

The semantic branch of the proposed Mask-PNet obtains a semantic segmentation result consisting of stuff and thing regions. After further processing in the mask MP branch, the thing segmentation regions are replaced by instance segmentations. Meanwhile, the stuff area logits are frozen as background logits. Due to the accurate prediction of the panoptic logits, the stuff regions are optimized, resulting in more accurate stuff areas.

We conducted an ablation study to demonstrate that the accuracy of stuff region improves in mIoU. The stuff classes of the Cityscapes dataset include 11 stuff classes, such as road, sidewalk, building, wall, fence, pole, traffic light, traffic sign, vegetation, terrain, and sky. As shown in Figure 4, the segmentation results of the stuff regions after the MP branch fused logits are more accurate than the segmentation results of the original semantic branch, and the accuracy is improved by about 8.1% on average compared to the weighted class regions. Improvements are mainly focused on the edge regions of the stuff class. Larger areas like the sky have smaller accuracy gains, while smaller areas like traffic signs have higher accuracy gains of up to 14.3%. However, for the stuff class pole, the recorded accuracy drops by 1.6%, a possible reason is that the pole class is too tall and thin (skinny shape). Overall, the MP branch significantly improves the segmentation accuracy of the stuff class, which confirms that the proposed branch has better accuracy than the original semantic segmentation branch.

### 4.3. Comparisons on Cityscapes Dataset

We compare the performance of Mask-PNet with several representative methods from different architectures, such as Weakly Supervised [34], Panoptic FPN [13], AUNet [28], UPSNet [6], Seamless [35], PCV [12], Mask RCNN [4], SSAP [36], AttentionPS [37], and MaskConver [38]. Table 1 shows the results of the Cityscapes validation set. The proposed Mask-PNet trained only on Cityscapes fine annotations with single-scale evaluation, without using the Cityscapes coarse annotations, depth data, or exploit temporal data.

Our Mask-PNet outperforms mainstream methods. For the PQ metric, our method outperforms the state-of-the-art UPSNet [6] by 1.98%, with a 3.85% lead in the “thing” class and a 1.02% lead in the ‘stuff’ class. For the average segmentation accuracy mIoU, our method outperforms the best AUNet [28] by 1.19%. Through the above analysis, we can find that our proposed Mask-PNet has the best recall rate RQ and has a competitive segmentation accuracy SQ.

### 4.4. Comparisons on COCO Dataset

In this sub-section, we compare the performance of our proposed Mask-PNet method with several existing state-of-the-art panoptic segmentation techniques. AUNet [28], Panoptic FPN [13], AdaptIS [37], UPSNet [6], Seamless [35], Mask RCNN [4], SSAP [36], AttentionPS [37], and MaskConver [38] are among the representative methods we compare against. To evaluate the performance of these methods, we utilize the COCO validation dataset and present the results in Table 2. Our proposed Mask-PNet demonstrates good performance and outperforms mainstream methods in terms of the Panoptic Quality (PQ) metric. Specifically, our method achieves a PQ score that is 0.2% higher than the state-of-the-art UPSNet [6]. Our comprehensive analysis of the results reveals that our Mask-PNet also achieves the highest Segmentation Quality (SQ) metric among all methods evaluated and exhibits competitive Recall Quality (RQ) scores.

These results provide strong evidence that our proposed method is effective in improving panoptic segmentation accuracy and outperforms the other listed techniques in terms of PQ, SQ, and RQ metrics on the COCO dataset.

### 4.5. Computing Resource Cost Comparison

Benefiting from the pyramid object detection mechanism, the proposed Mask-PNet aims to perform segmentation in a faster and more efficient manner. We compared our Mask-PNet with Mask RCNN [4], the most famous box-based segmentation architecture. The forward pass FLOPs (Floating-point operations per second) and the model parameters cost are measured and reported. Both Mask-PNet and Mask RCNN are tested on the ResNet-50 backbone version. For each forward pass, the same group of 60 random rich objects 512 × 512 images are fed into the model networks, then the average GPU memory cost is measured. The measurement results are listed in Table 3.

From the statistics in Table 3, compared with Mask RCNN, the proposed Mask Pyramids Network obtains a 5.2% higher mIoU performance while costing 14% less FLOPs and 17.6% less GPU memory resources. In order not to miss small objects, box-based methods Mask RCNN tend to filter and reserve more box proposals, which increases the burden of the post-segmentation process. On the contrary, Mask-PNet avoids generating a large number of box proposals and costs fewer resources by a large margin. We measure the proposal generating number of Mask RCNN and proposed Mask-PNet, and for a 512*512*3 input image, Mask RCNN generates 65,472 proposals and selects 1000 highest-score proposals for segmentation. Our Mask-PNet generates 951 proposals on average in the 2014val folder of Cityscapes, the proposal number varies from different images. By comparison, Mask-PNet can reduce the generated proposals by a large margin, significantly reducing the computation burden. When compared with more recent work by MaskConver, our model is not as fast as MaskConver, but our model costs less computation resources, i.e., the FLOPs cost of ours is only one-third of that of MaskConver. This is mainly because our model spends more time rounds to predict the segmentation results, which makes it possible to perform faster on smaller devices.

In the future, we will continue to improve the Mask-PNet framework. We will explore the parallel algorithm of this framework in object detection applications. This is because the conventional approach cascades the detection and segmentation of objects from large to small, which slows down the overall segmentation speed. At the same time, we also hope to extend the Mask-PNet architecture from image processing to video segmentation and understanding.

### 4.6. Visualization

The panoptic segmentation results of Mask RCNN are compared with the proposed Mask-PNet as shown in Figure 5. It is not difficult to determine from the visualization that the segmentation edges of Mask-PNet are sharper and more accurate. Our Mask-PNet can segment both ‘thing’ and ‘stuff’ areas very accurately. We also provide the COCO validation dataset performance visualization results in Figure 6.

## 5. Conclusions

We propose a novel panoptic segmentation method called the Mask-Pyramid Network (Mask-PNet). Instead of densely generating a large number of box proposals and filtering, we advocate exploiting coverage and the suppression of existing masks to avoid generating too many useless box proposals. The proposed Mask-PNet adopts a new pyramid proposal generation mechanism that significantly reduces the number of proposals and thus greatly reduces the computational complexity of the inference process. In addition, the proposed method produces a unified masked logits tensor and merges semantic and instance segmentation results interactively, which naturally balances the masked logits values from the two segmentation subtasks. Experimental results show that our Mask-PNet can achieve 2% higher accuracy than the state-of-the-art panoptic segmentation method while requiring much fewer computational resources.

## Figures and Tables

**Figure 1 sensors-24-01411-f001:**
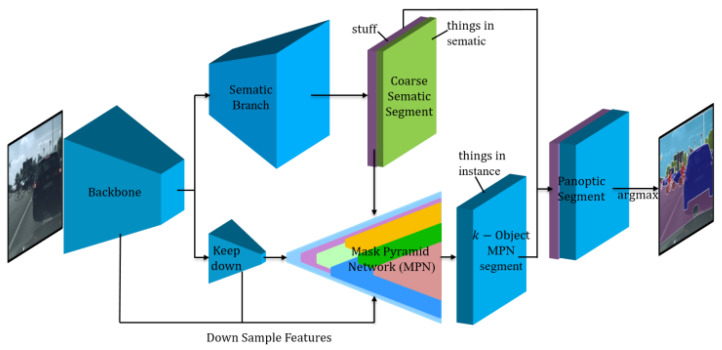
Overview of Mask-Pyramid Network architecture. It consists of Semantic Branch and Mask-Pyramid Branch. Semantic Branch produces coarse semantic segmentation mask logits, the segmentation mask will be improved when calculating SoftMax with MPN segmentation logits later. Mask-Pyramid Branch generates mask logits for each instance object in the image. Both branches generate H*W*k segmentation logits, the concatenated logits are naturally fused by applying SoftMax operation.

**Figure 2 sensors-24-01411-f002:**
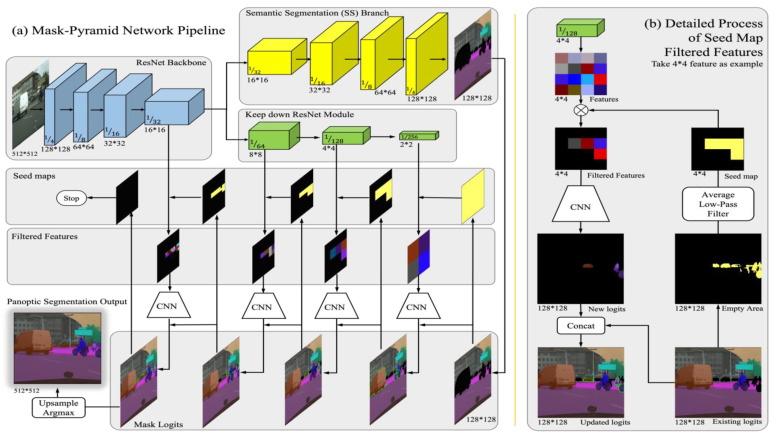
Overview of the Mask-PNet Pipeline. The left part (**a**) shows the overall interactive flow. Take the “stuff” layers of the semantic segmentation result as the initial mask logits. Then, generate a seed map based on the empty area of the mask logits, and filter the ResNet features with the seed map. Afterward, feed the filtered features into CNN to obtain logits of the newly detected objects, and merge new logits to update the mask logits. Next, generate a new seed map and repeat these steps until the seed map is fulfilled. The right part (**b**) demonstrates in detail the process of generating a seed map and filtered features. Take the 4*4 feature level as an example. First, extract the empty area of the existing logits. By using the average low-pass filter, we can downsample the 128*128 empty area map into a 4*4 Boolean seed map. Then, elementary multiply the seed map with the features from the ResNet to obtain the filtered feature. Next, generate new logits via CNN and concatenate them with existing mask logits to obtain updated mask logits and execute the next loop.

**Figure 3 sensors-24-01411-f003:**
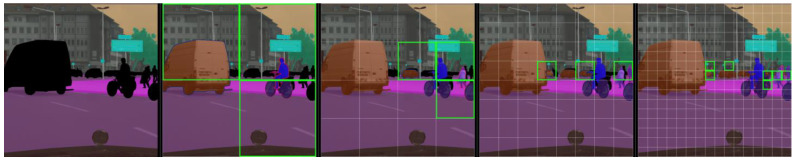
Illustration of the mask logits updating process. The first image shows the initial mask logits from the “stuff” area of the semantic segmentation result. The second image shows three newly generated object masks from the 2*2 features, the seed map provides four proposals, CNN predicts three valid mask logits, and the other logit is suppressed by the “stuff” logits. The third and fourth images show three new mask logits generated from the 4*4, and 8*8 features, respectively. The last image shows seven newly generated mask logits out of the 16*16 features.

**Figure 4 sensors-24-01411-f004:**
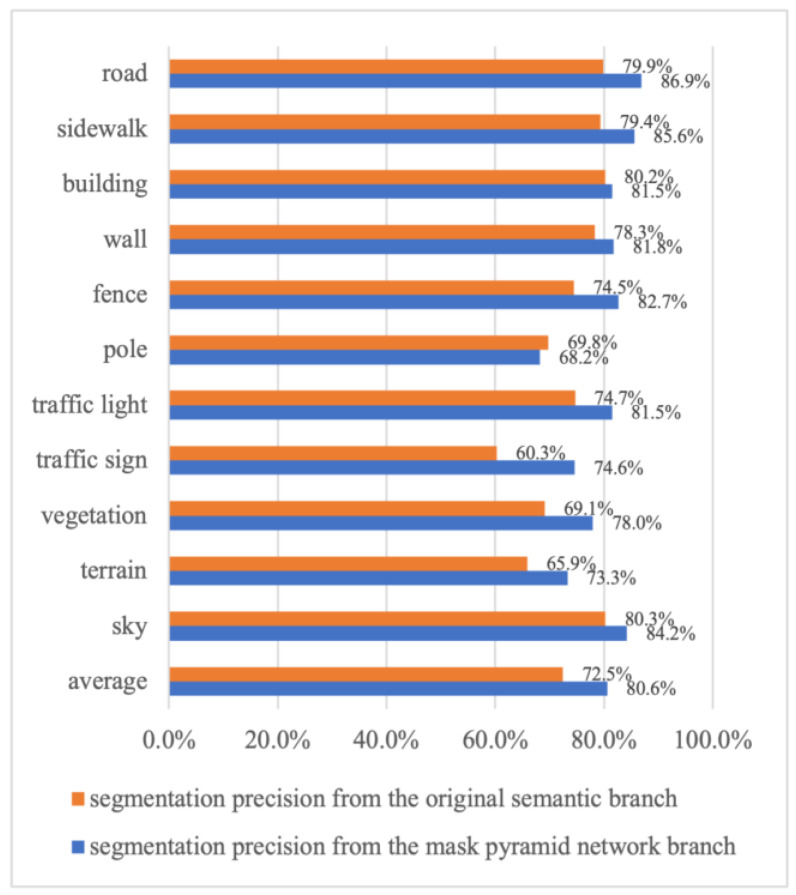
The segmentation precision comparison between the semantic segmentation branch result and the final fused result on “stuff” categories. At all categories, the “stuff” segmentation results after the MP branch fused logits are more accurate than the results of the original semantic branch.

**Figure 5 sensors-24-01411-f005:**
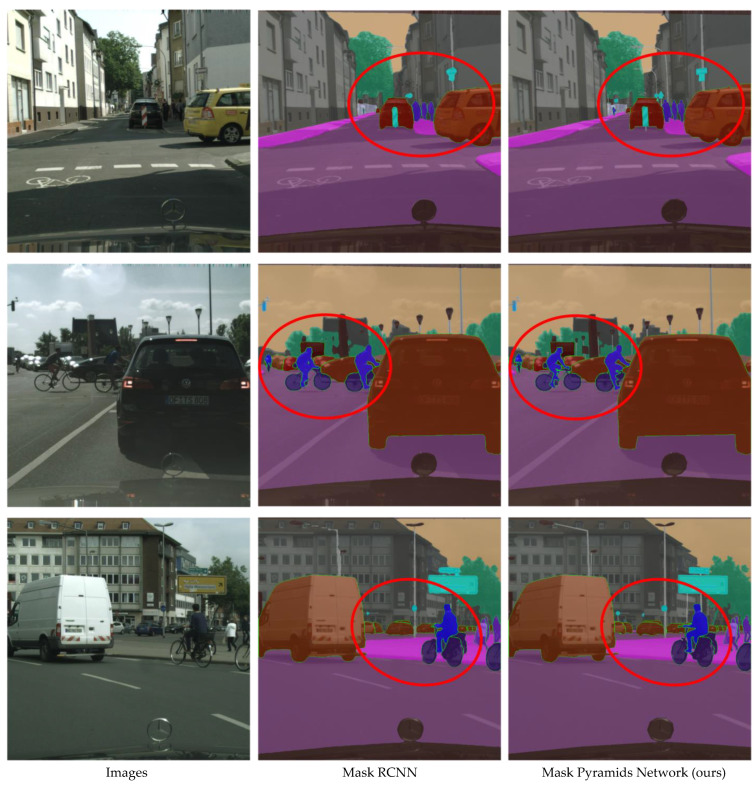
Visualization of panoptic segmentation performance comparison between Mask RCNN and proposed Mask Pyramids Network on the Cityscapes validation set. The difference area is marked in red circles.

**Figure 6 sensors-24-01411-f006:**
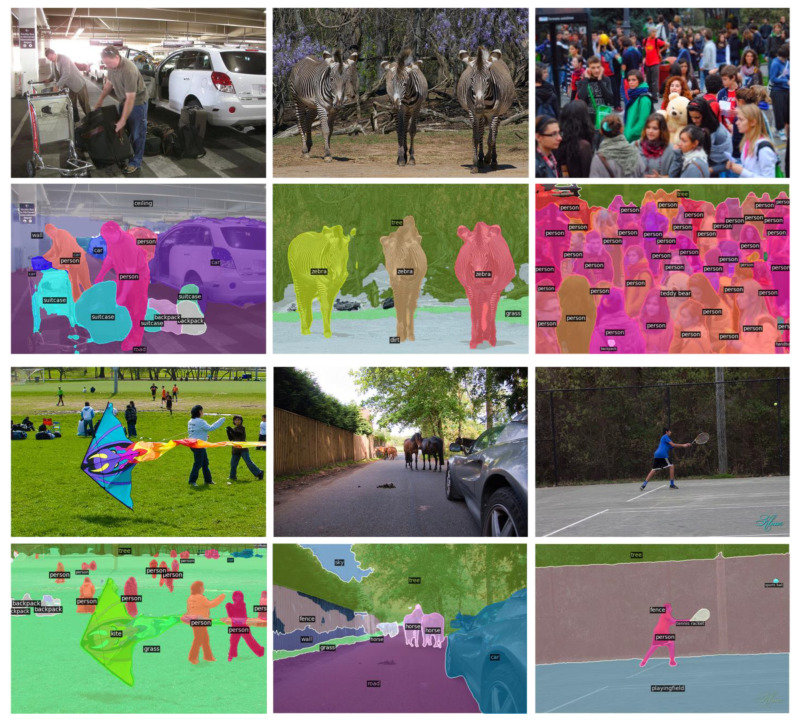
Visualization of panoptic segmentation performance results of the Mask Pyramids Network on the COCO validation dataset.

**Table 1 sensors-24-01411-t001:** Panoptic segmentation performance comparison on the Cityscapes validation dataset. Superscripts St and Th refer to ‘stuff’ and ‘thing’ classes, respectively.

Methods	PQ (%)	SQ (%)	RQ (%)	PQ^Th^ (%)	SQ^Th^ (%)	RQ^Th^ (%)	PQ^St^ (%)	SQ^St^ (%)	RQ^St^ (%)	mIoU (%)
Weakly Supervised	47.3	61.9	76.4	39.6	62.4	63.5	52.9	68.0	77.8	65.6
Panoptic FPN	58.1	85.0	68.4	52.0	79.7	64.6	61.5	87.4	69.5	70.7
AUNet	59.1	85.8	68.9	54.8	81.3	66.8	62.1	89.7	67.9	75.6
UPSNet	59.5	85.8	69.3	57.2	83.3	68.7	62.7	80.1	76.2	74.5
Seamless	60.2	87.7	69.0	55.1	82.2	66.3	63.3	89.9	69.2	79.0
PCV	54.2	69.9	77.5	47.8	70.6	67.7	58.9	79.2	74.4	74.1
Mask RCNN	53.9	68.3	78.9	50.5	65.7	76.9	56.4	71.1	79.3	71.3
AttentionPS	59.7	-	-	52.8	-	-	64.1	-	-	76.0
MaskConver	53.6	-	-	58.9	-	-	45.6	-	-	76.4
SSAP	61.0	88.5	68.9	55.0	86.0	62.3	60.2	88.9	77.3	72.8
**Mask-PNet**	**61.8**	79.7	77.4	**59.4**	76.1	78.1	**64.0**	80.5	79.8	**79.8**

**Table 2 sensors-24-01411-t002:** Panoptic Segmentation Performance Comparison on the COCO Validation Dataset.

Methods	PQ (%)	SQ (%)	RQ (%)
AUNet	35.5	78.0	45.4
Panoptic-FPN	36.1	79.3	45.5
AdaptIS	36.9	82.1	44.9
UPSNet	37.1	82.2	45.1
Seamless	37.2	80.6	**46.2**
Mask RCNN	31.8	70.7	45.0
SSAP	32.5	73.3	44.3
AttentionPS	34.4	-	-
MaskConver	37.2	-	-
**Mask-PNet**	**37.3**	**82.4**	45.6

**Table 3 sensors-24-01411-t003:** Performance and Resource Consumption Comparison between Mask RCNN and proposed Mask Pyramids Network.

Methods	PQ↑	Proposal Number↓	GFLOPs↓	GPU Mem Cost↓ (MB)	FPS ↑
Mask RCNN	31.8	65,472	269.544	3072	32
MaskConver	37.2	-	696	-	**105**
**Mask-PNet**	**37.3**	951	**231.806**	**2530**	46

## Data Availability

The data presented in this study are openly available in Cityscapes at https://doi.org/10.1109/CVPR.2016.350, reference number [32].

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
