# Peer review of "Mask-Pyramid Network: A Novel Panoptic Segmentation Method"

_sensors, 2024, doi:10.3390/s24051411_

Round 1

Reviewer 1 Report

Comments and Suggestions for Authors

This paper proposed a new mask pyramid mechanism to distinguish objects and generate much fewer proposals by referring existing segmented masks, so as to reduce computing resource consumption. The paper is organized and written well in English, all technical issues are correct, extensive experimental results are convincing. This reviewer has the following concerns.

(1)   More details about the semantic branch and coarse sematic segment modules in Fig 1 are suggested to be addressed, so that the readers can get more useful information.

(2)   Dose the semantic segmentation results obtained by the semantic branch have a strong influence on the final results?

(3)   For Eq. (1) and (2),the definition of variables should be addressed before using it. For instance, How are the index i and j related to thing and logits? What is the meaning of yi and pi?

(4)   More newly proposed methods (proposed in 2022 and 2023) should be compared to show the effectiveness of the proposed method.

Comments on the Quality of English Language

Minor editing of English language required

Author Response

Point-by-point response to Comments and Suggestions for Authors.  

Comments 1: More details about the semantic branch and coarse semantic segment modules in Fig 1 are suggested to be addressed, so that the readers can get more useful information. Response 1: Thank you for pointing this out. We agree with this comment. Therefore, we have added more details about the semantic branch and coarse semantic segment modules in Fig. 1, which says " Semantic Branch produces coarse semantic segmentation mask logits, the segmentation mask will be improved when calculating SoftMax with MPN segmentation logits later. Mask-Pyramid Branch generates mask logits for each instance object in the image."  

Comments 2: Dose the semantic segmentation results obtained by the semantic branch have a strong influence on the final results? Response 2: Yes. Semantic segmentation affects the final results in two ways. On the one hand, the coarse segmentation result of semantic branch is an important basis for dividing things and stuff. On the other hand, the logits of stuff in the semantic segmentation result will be SoftMaxed together with the logits of instance object to produce the final segmentation result. In order to illustrate this impact more clearly, we further describe it in section 3.1.  

Comments 3: For Eq. (1) and (2),the definition of variables should be addressed before using it. For instance, How are the index i and j related to thing and logits? What is the meaning of yi and pi? Response 3: Thanks a lot for pointing this out. We have added the detailed definition of variables in lines 353-355, 385-388. Which says, "As is expressed in Equation 1, the pixel value of the seed map at the coordinate , is marked as 1 if the corresponding value of things map is larger than the logits value, otherwise, it will be marked as 0.", "where i refers to each pixel, p_i refers to the pixel of the predicted mask, y_i refers to the pixel of the ground truth label."  

Comments 4: More newly proposed methods (proposed in 2022 and 2023) should be compared to show the effectiveness of the proposed method. Response 4: Thanks for the good proposal. We reviewed 2 more recent works from 2022  to 2024, covering from MDPI Sensors to WACV to further compare our proposed model. The results shows we can also provide comparable resultsWhen compared with more recent works, our model is not as fast as MaskConver, but our model costs less computation resources, i.e., the FLOPs cost of ours is only 1/3 of the MaskConver. This is mainly because our model spends more time rounds to predict the segmentation results, which make it possible to perform faster on smaller devices. Limited by only giving us 10 days to revise, we were unable to reproduce the new work, so we directly used the metrics results reflected in their papers. Limited by only giving us 10 days to revise, we were unable to reproduce the new works, so we directly used the metrics results in their papers. Relevant changes have been made in many places in the article and are highlighted.

Reviewer 2 Report

Comments and Suggestions for Authors

This paper proposes a mask-pyramid network and a joint training mechanism for panoptic segmentation. This paper feeds the output features of a branch CNN network with masks generated by the semantic segmentation branch to mask logits gradually, aiming to recognize objects from big to small. Furthermore, the authors conduct the experiments on the Cityscape and COCO dataset. There are some concerns that need to be addressed.

1. Contributions are not explained clearly. In Line 134, the authors propose a novel joint training mechanism. Please explain more about how it works and point it out clearly in the proposed method of the paper.

2. This paper only compares the methods in 2021. Please supplement the comparison experiments with methods in 2022 and 2023.

3. This paper only compares the computing resource cost of the proposed method with one network. Please supplement more comparison experiments on computing resource cost with other networks.

4. Please point out ground truth, Mask RCNN, and the proposed method in Figure 5.

Comments on the Quality of English Language

Moderate editing of English language required.

Author Response

Point-by-point response to Comments and Suggestions for Authors.  

Comments 1: Contributions are not explained clearly. In Line 134, the authors propose a novel joint training mechanism. Please explain more about how it works and point it out clearly in the proposed method of the paper.

Response 1:  Thank you for pointing this out. We agree with this comment. Therefore, we have added more details about the joint training mechanism that enables the interaction of gradients and loss information. We added the description in line 218-222 and 235-239. Which says "In this way, the k-object mask logits and the stuff mask logits are combined together to conduct the cross-entropy loss, the loss function can directly calculate the gradients based on the overview of semantic segmentation branch and the mask pyramid branch. This provides a novel joint training mechanism that enables the interaction of gradients and loss information between the semantic and instance segmentation subtasks.", "Semantic segmentation branch affects the final results in two ways. On the one hand, the coarse segmentation result of semantic branch is an important basis for dividing things and stuff. On the other hand, the logits of stuff in the semantic segmentation result will be concatenated together with the logits of instance object, so as to conduct SoftMax for producing the final segmentation result."

Comments 2: This paper only compares the methods in 2021. Please supplement the comparison experiments with methods in 2022 and 2023.

Response 2:  Thanks for the good proposal. We added 2 more recent works from 2022  to 2024, covering from MDPI Sensors to WACV to further compare our proposed model. The results shows we can also provide comparable results. When compared with more recent works, our model is not as fast as MaskConver, but our model costs less computation resources, i.e., the FLOPs cost of ours is only 1/3 of the MaskConver. This is mainly because our model spends more time rounds to predict the segmentation results, which make it possible to perform faster on smaller devices. Limited by only giving us 10 days to revise, we were unable to reproduce the new work, so we directly used the metrics results reflected in their papers. Limited by only giving us 10 days to revise, we were unable to reproduce the new works, so we directly used the metrics results in their papers. Relevant changes have been made in many places in the article and are highlighted.

Comments 3: This paper only compares the computing resource cost of the proposed method with one network. Please supplement more comparison experiments on computing resource cost with other networks.

Response 3:  Thanks a lot for indicating this problem. We investigated on several  related papers, find that only few panoptic segmentation papers report their computing resource information in details. We added the computing resource infromation from the paper of MaskConver into the comparison in Table 3, and find that when compared with more recent works, our model is not as fast as MaskConver, but our model costs less computation resources, i.e., the FLOPs cost of ours is only 1/3 of the MaskConver. This is mainly because our model spends more time rounds to predict the segmentation results, which make it possible to perform faster on smaller devices. Limited by only giving us 10 days to revise, we were unable to reproduce the new work, so we directly used the metrics results reflected in their papers. Limited by only giving us 10 days to revise, we were unable to reproduce the new works, so we directly used the metrics results in their papers. Relevant changes have been made in many places in the article and are highlighted. The relative finding are updated in line 510-515 of the manuscript.

Comments 4: Please point out ground truth, Mask RCNN, and the proposed method in Figure 5.

Response 4:  Many thanks for pointing this out. We agree with this comment and added  the ground truth, Mask RCNN, and the proposed method information in the Figure 5.

Round 2

Reviewer 2 Report

Comments and Suggestions for Authors

This paper can be accepted.

Comments on the Quality of English Language

None.

Author Response

Thank you for your suggestions. We further polished the introduction, abstract, related works and conclusion chapters to make it easier for readers to read.